

# Overlapping community discovery based on graph embedding and label propagation algorithm

Miaomiao Liu[1,2], Yuchen Liu[1], Yanan Hu[1], Jing Chen[3] and Wenqing Zhang[1]

[1] School of Computer and Information Technology, Northeast Petroleum University, Daqing, Heilongjiang, China
[2] SanYa Offshore Oil & Gas Research Institute, Northeast Petroleum University, Sanya, Hainan, China
[3] School of Mathematics and Computer Science, Guangdong Ocean University, Zhanjiang, Guangdong, China

## ABSTRACT

Traditional label propagation algorithms (LPA) exhibit instability and poor accuracy in community discovery, primarily due to random node selection, uncertain label update sequences, and neglect of node importance variations. We present GELPA-OCD (overlapping community discovery based on graph embedding and label propagation algorithm), an overlapping community discovery algorithm that integrates graph embedding with label propagation to address these limitations. Our approach introduces a multidimensional node importance assessment strategy and employs Node2vec graph embedding to represent nodes as low-dimensional vectors, effectively capturing network structure features. The algorithm employs similarity-based weight factors to guide label propagation and implements adaptive filtering mechanisms to enhance effectiveness. We conduct experiments on both real and artificial datasets. Using *EQ, NMI*, and F1-score as evaluation metrics, the experimental results show that the proposed algorithm effectively reduces randomness and uncertainty in node selection and label updating processes, achieving more stable and accurate overlapping community discovery.

## INTRODUCTION

### Research background

With the rapid advancement of Internet technologies, complex network analysis has become increasingly interdisciplinary, spanning information science and biology, and establishing itself as a research field of considerable theoretical and practical significance (*Feng et al., 2021*). As an important feature of complex networks, community structure refers to the phenomenon where nodes within the same community are densely connected, while connections between nodes in different communities are relatively sparse (*Liu et al., 2024*). Traditional non-overlapping community discovery methods divide complex networks into several communities, ensuring that each node belongs to only one

Corresponding author
Yuchen Liu, lyc02241118@163.com

community. However, in real-world networks, nodes often belong to multiple communities, and these overlapping nodes are of great significance for understanding the topology and functionality of the entire network.

Nowadays, the label propagation algorithms (LPA)-based approach has gained widespread adoption in community detection due to its computational simplicity and near-linear time complexity (*Chen, Jiang & Guo, 2023*; *Wang et al., 2023*). Therefore, improving the stability of LPA-based overlapping community detection algorithms has become a key issue.

## Related work

Overlapping community structures have attracted considerable attention as they more accurately reflect real-world network characteristics (*Zhuo et al., 2024*). Consequently, researchers have developed numerous algorithms for overlapping community discovery, each tailored to specific methodological approaches and network topologies (*Feng et al., 2024*). The Clique Percolation Method (CPM) (*Palla & Farkas, 2005*) is a representative algorithm for overlapping community discovery that detects communities by identifying k-cliques and their overlaps in the network. However, its performance is sensitive to the value of $k$, thus limiting its applicability to large-scale or sparse networks. Local Fitness Maximization (LFM) (*Lancichinetti, Fortunato & Kertész, 2009*) algorithm detects overlapping communities by optimizing a local fitness function that measures the strength of community structure around each node, but it suffers from instability due to random seed selection.

Traditional Label Propagation Algorithm (LPA) algorithms are unstable and limited to non-overlapping community detection (*Li et al., 2022a*). *Gregory (2010)* addressed this limitation by proposing the Community Overlap Propagation Algorithm (COPRA), which extends label propagation to overlapping community discovery by allowing nodes to belong to multiple communities with membership coefficients. The COPRA algorithm offers the advantages of low time complexity and suitability for large-scale networks. However, the order of node updates and label selection remains highly random, which results in unstable community discovery results. *Lu et al. (2018)* proposed the LPA with Neighbor Node Influence (LPANNI) algorithm, which incorporates node influence measures to guide the label propagation process. This algorithm overcame the limitations of traditional methods that ignored the importance of nodes. *El Kouni, Karoui & Ben Romdhane (2020)* extended the node importance measurement method by incorporating additional topological features. *Wang et al. (2022)* proposed a node-label overlapping community partition algorithm based on entropy transformation that uses information entropy to optimize community assignments. *Tang, Li & Tang (2021)* proposed the Detecting Overlapping Community based on LPA (DOCLPA) algorithm to address the randomness issue by introducing deterministic node ordering strategies. However, this algorithm is not suitable for sparse networks.

Recent advances in overlapping community detection have explored several promising directions. First, graph embedding integration has attracted increasing research attention, with studies investigating the combination of network representation learning with community detection methods. *Yang, Wang & Ye (2022)* explored this approach by

integrating the Node2vec model (*Grover & Leskovec, 2016*) with community detection based on node similarity. *Chen et al. (2020)* developed this direction further with their Community Detection *via* Deep Variational Gaussian Mixture (CD-DVG) algorithm, which employs a two-stage strategy that first extracts node features using deep learning techniques and then identifies community structures through vector graph embedding methods. *Hu et al. (2020)* investigated this integration by combining Node2vec with spectral clustering methods, though their approach requires specifying the number of communities in advance and has high computational complexity. Second, multi-modal feature learning represents another research direction that addresses the limitations of purely topological approaches. *Berahmand et al. (2024)* proposed a semi-supervised deep attribute clustering method using dual auto-encoders that can simultaneously learn both structural and attribute features of networks. While these auto-encoder approaches can capture nonlinear data representations more effectively than Node2vec, they are less efficient when processing pure topological structures. *Berahmand et al. (2025)* provided a comprehensive review of research that integrates graph structure learning into spectral clustering, establishing a theoretical foundation for understanding the global structural features of networks. However, spectral clustering methods typically have higher computational complexity. Third, higher-order relationship modeling has received attention as researchers seek to capture complex node interactions beyond simple pairwise relationships. *Sheikhpour et al. (2025)* developed semi-supervised discriminant analysis methods using hypergraph Laplacian structures, which enable the capture of higher-order relationships among multiple nodes through hypergraph representations. Despite these advances, current methods face various challenges: local information dependency that may neglect global network influence (*Gao et al., 2024*), computational complexity trade-offs, and stability issues in overlapping scenarios. These challenges motivate the development of approaches that can balance embedding effectiveness with computational efficiency while maintaining stability in overlapping community detection.

## Contribution of the study

This article proposes GELPA-OCD (overlapping community discovery based on graph embedding and label propagation algorithm), a method based on graph embedding and label propagation algorithm for overlapping community discovery. This study's main contributions are as follows.

(1) We propose a node importance measurement index that combines degree centrality, PageRank value, and local clustering coefficient. Based on the descending order of node importance values, we determine the update sequence of node labels, effectively reducing the randomness in node selection and improving the stability of community discovery results.

(2) In the label propagation strategy, we integrate Node2vec graph embedding to construct node similarity matrices. Based on this, we propose a new method for computing the label attribution coefficient. Moreover, by setting an adaptive threshold for label filtering, the randomness and computational cost of label selection are effectively reduced, improving the algorithm's stability, accuracy, and efficiency.

(3) We comprehensively evaluated the proposed algorithm using multiple metrics (EQ, NMI, F1-score) on six real-world networks and 25 artificial datasets, demonstrating its outstanding performance in overlapping community detection.

## THEORETICAL BASIS

### Graph embedding

This study focuses on community discovery in undirected and unweighted networks, which are typically represented as a graph $G(V, E)$, where $V$ is the node set and $E$ is the edge set of the network. Graph embedding is a technique that maps nodes, edges, or subgraphs in a given graph structure to a low-dimensional vector space. Representative algorithms for graph embedding models include Deepwalk, Node2vec (*Johnson, Murty & Navakanth, 2024*), and others. *Grover & Leskovec (2016)* improved the random walk method based on Deepwalk and proposed the Node2vec algorithm. Unlike the Deepwalk algorithm, Node2vec incorporates Breadth First Search (BFS) and Depth First Search (DFS) into random walks. It controls the direction of random walks through parameters $p$ and $q$ to obtain richer network structure information (*Zhang, 2024*). In this study, we use the Node2vec model to learn the topological structure of the network. Combining Node2vec with overlapping community discovery can considerably preserve the topology information of the network and ensure the accuracy of overlapping community discovery results (*Gao et al., 2024*).

### Related concepts and definitions

The following briefly explains the relevant concepts and definitions involved in the algorithm proposed in this article. In the following text, $N$ represents the total number of nodes in the network, $d_v$ represents the degree of node $v$, $\Gamma(v)$ represents the set of neighboring nodes of node $v$, and '$\|$' represents the number of elements in the set.

**Definition 1: Degree centrality** (*DC*). The degree of centrality measures the direct influence of a node and evaluates its importance. We denote the degree centrality of node $v$ as $DC(v)$, which is defined as in Formula (1).

$$DC(v) = \frac{d_v}{N-1}.\tag{1}$$

**Definition 2: PageRank value** (*PR*). The PageRank value reflects the global importance of a node in the network. We denote the PageRank value of node $v$ as $PR(v)$, which is defined as in Formula (2), where $\alpha$ is the damping coefficient set to 0.85.

$$PR(v) = \frac{(1-\alpha)}{N} + \alpha \sum_{v' \epsilon \Gamma(v)} \frac{PR(v')}{d_{v'}}.\tag{2}$$

**Definition 3: Clustering coefficients** (*CC*). The clustering coefficient of a node describes the density of edges around the node. We denote the clustering coefficient of node $v$ as
$CC(v)$, which is defined as in Formula (3), where $T_v$ represents the number of edges between neighboring nodes $v$.

$$CC(v) = \frac{2T_v}{d_v \times (d_v - 1)}.$$ (3)

## GELPA-OCD METHODS

LPA-based overlapping community discovery algorithms suffer from substantial randomness in node update ordering and label selection, leading to unstable and inaccurate community detection results. To address these issues, this article introduces the GELPA-OCD algorithm for overlapping community detection, which integrates graph embedding and LPA. We construct a node importance evaluation index in the label initialization stage by comprehensively considering nodes' local structural characteristics and global topological features. We then determine the update order of node labels based on the descending order of this importance index, aiming to address the instability of community discovery results caused by the random selection of node label update order. In the label propagation stage, the algorithm introduces the graph embedding model Node2vec to preprocess the network, constructs a similarity matrix, and integrates it into calculating improved label attribution coefficients, thus proposing a cosine similarity-based label attribution coefficient. Due to the differences in similarity between each pair of nodes, the method can effectively avoid the situation where nodes have the same label attribution coefficient in community attribution determination, reducing the probability of nodes randomly selecting labels, and improving the accuracy and stability of the algorithm for community discovery results.

### Improvement of the network representation preprocessing stage

The Node2vec algorithm can balance the maintenance of local structural features and the capture of global network information by adjusting parameters $p$ and $q$ for complex networks. Additionally, Node2vec's random walk strategy is more flexible and can adjust sampling preferences according to actual needs, thereby obtaining more targeted node representations. The algorithm proposed in this article uses Node2vec to learn the network topology representation, mapping nodes to a low-dimensional vector space. We use cosine similarity to measure the similarity between vectors. Its calculation method is the product of the inner product of two vectors divided by the vector modulus, as shown in Formula (4).

$$sim(i, j) = \frac{f_i \cdot f_j}{\|f_i\| \cdot \|f_j\|} = \frac{\sum_{d=1}^{D} f_{(i,d)} \cdot f_{(j,d)}}{\sqrt{\left[\sum_{d=1}^{D} f_{(i,d)}^2\right]} \sqrt{\left[\sum_{d=1}^{D} f_{(j,d)}^2\right]}}.$$ (4)

In Formula (4), $sim(i, j)$ represents the cosine similarity between node $i$ and node $j$, $f_i$ represents the embedding vector of node $i$, $f_{(i,d)}$ represents the $d$-th dimensional component of the embedding vector of node $i$, and $D$ is the dimension of the embedding vector. Based on Formula (4), we can construct a node similarity matrix $S$ and apply it to the subsequent label propagation process.

### Improvement in the label initialization phase

*Node importance evaluation integrating multidimensional indicators*

When evaluating the importance of a node in overlapping community discovery algorithms, existing research mainly uses a single indicator, such as degree centrality, to measure it (*Liu et al., 2020*). Given this, we propose a node importance evaluation indicator integrating multidimensional features, comprehensively considering three dimensions: degree centrality, PageRank value, and the clustering coefficient of nodes. First, degree centrality measures the direct connection relationship between nodes and other nodes. Second, the PageRank value evaluates the influence of nodes on the entire network. Finally, the clustering coefficient measures the degree of connectivity between neighbors of a node. After normalization, we fuse these three indicators to avoid the potential one-sidedness of a single indicator. In this article, the importance of node $v$ is denoted as $NI(v)$, as shown in Formula (5). Here, $DC\_norm(v)$, $PR\_norm(v)$, and $CC\_norm(v)$ represent the normalized degree centrality, normalized PageRank value, and normalized clustering coefficient of node $v$, respectively. Their calculation methods are shown in Formulas (6), (7), and (8), respectively.

$$NI(v) = DC\_norm(v) \times PR\_norm(v) \times (1 + CC\_norm(v)). \tag{5}$$

$$DC\_norm(v) = \frac{DC(v)}{max_{u \in V} DC(u)}. \tag{6}$$

$$PR\_norm(v) = \frac{PR(v)}{max_{u \in V} PR(u)}. \tag{7}$$

$$CC\_norm(v) = \frac{CC(v)}{max_{u \in V} CC(u)}. \tag{8}$$

Here, $DC(v)$, $PR(v)$, and $CC(v)$ are the degree centrality, PageRank value, and clustering coefficient of node $v$ defined in Formulas (1), (2), and (3), respectively. $Max()$ represents taking the maximum value of the corresponding indicator.

*The strategy of label initialization*

Based on the above node importance evaluation method, this article adopts the following strategy for label initialization. Firstly, we calculate the importance values of all nodes in the network. Secondly, we sort them in descending order to form an updated sequence of node labels. Finally, for each node, we assign a unique initial label and set the attribution coefficient of its initial label to 1. The pseudocode for the node importance-based label initialization strategy is presented as follows.

### Improvement in the label propagation stage

The traditional LPA-based community discovery method uses a single indicator for calculating the label attribution coefficient, which fails to measure the community to which the node belongs effectively. This article proposes an improved label propagation mechanism to address the aforementioned issues.

| Algorithm Label initialization strategy based on node importance. |
|---|
| Input: $G(V, E)$ |
| Output: update_sequence, initialized_labels |
| Step 1: Calculate basic indicators |
|     Initialize update_sequence =[ ] |
|     For each $v \in V$: Calculate $DC(v)$ according to Formula (1); End For |
|     For each $v \in V$: Calculate $PR(v)$ according to Formula (2); End For |
|     For each $v \in V$: Calculate $CC(v)$ according to Formula (3); End For |
| Step 2: Calculate node importance |
|     For each $v \in V$: Calculate $DC\_norm(v)$ according to Formula (8); End For |
|     For each $v \in V$: Calculate $PR\_norm(v)$ according to Formula (9); End For |
|     For each $v \in V$: Calculate $CC\_norm(v)$ according to Formula (10); End For |
|     For each $v \in V$: Calculate $NI(v)$ according to Formula (7); End For |
| Step 3: Generate update sequence |
|     sorted_nodes = Sort $(V)$ by $NI(v)$ in descending order |
|     For each v $\in$ sorted_nodes: update_sequence.append($v$) ; End For |
| Step 4: Initialize labels |
|     initialized_labels = { } |
|     For each $v \in$ update_sequence |
|     set initialized labels of v to 1; initialized_labels=initialized_labels$\cup\{(v,1)\}$ |
|     End For |
|     Return update_sequence, initialized_labels |

### *Label update strategy based on node attribution coefficient*

When nodes receive multiple candidate labels with identical attribution coefficients during filtering, random label selection compromises algorithm stability (Li et al., 2022b). To reduce the instability and low accuracy of community discovery results caused by the randomness of label selection, the algorithm proposed in this article introduces the Node2vec graph embedding technique. Our method uses cosine similarity to construct a node similarity matrix and introduces a novel Label Membership Coefficient ($LMC$). This coefficient not only considers the label weights of neighboring nodes but also introduces node similarity based on Node2vec as a weight factor, which enables the label propagation process to reflect the strength of relationships between nodes more accurately.

The label membership coefficient of a node reflects the degree to which that node belongs to a community. The membership coefficient of node $v$ to label $l$ is denoted as $LMC(v, l)$, which is defined as Formula (9), where $L(v)$ represents the label set of node $v$. Here, $L(v)$ represents the set of neighboring labels of node $v$, defined as shown in Formula (10).

$$LMC(v, l) = \frac{\sum_{u \in \Gamma(v)} sim(v, u) \times LMC(u, l)}{\sum_{u \in \Gamma(v)} \sum_{l' \in L(u)} sim(v, u) \times LMC(u, l')}. \tag{9}$$

$$L(v) = \{(l, LMC(u, l)) \mid u \in \Gamma(v)\}. \tag{10}$$

### *Rules of adaptive label filtering*

This article introduces an adaptive label filtering mechanism to ensure the effectiveness of label propagation. For node $v$, we set a label filtering threshold $\theta = \frac{1}{|L(v)|}$. Here, $|L(v)|$ represents the number of labels received by node $v$. If the attribution coefficient $LMC(v, l$

of a particular label $l$ to node $v$ is lower than the threshold $\theta$, we delete the label. If a node retains no neighbor labels after filtering, it keeps the neighbor label with the highest attribution coefficient.

### Label propagation process

The specific rules for the label propagation process are as follows.

(1) Data preprocessing. Using the Node2vec model to generate embedding vectors for nodes, calculate the cosine similarity between nodes based on Formula (4) and construct the inter-node similarity matrix $S$ based on this.

(2) Community label initialization. For each node $v$ in network graph $G$, calculate each node's importance index $NI(v)$ according to Formula (5), sort all nodes in descending order according to their $NI$ values, and form an updated sequence of node labels, denoted as update_sequence. Setting each node's initial label attribution coefficient $LMC(v, l)$ is 1.

(3) Label collection. Collect the label information of neighboring nodes for node $v$ to be updated in updates_sequence and form a neighbor label set $L(v)$ according to Formula (10).

(4) Calculation of attribution coefficient. Based on the similarity matrix $S$ and the label set $L(v)$ of nodes, calculate the new attribution coefficient $LMC(v, l)$ of node $v$ to label $l$ according to Formula (9).

(5) Label filtering. Filter the label attribution coefficient of node $v$ according to the set threshold $\theta$, and delete labels below the threshold. If no labels remain after filtering, the algorithm retains the label with the highest attribution coefficient.

(6) Normalization processing. Normalize the attribution coefficients in the retained label set so that the sum of the attribution coefficients for all labels is 1, resulting in a new label set $L'(v)$ for node $v$.

(7) Iteration until termination. Repeat steps (3) to (6), with one iteration per round. When the algorithm reaches the maximum number of iterations or the label attribution coefficients of all nodes remain unchanged between two consecutive iterations, the algorithm ends. Ultimately, all labels retained by each node represent the community to which it belongs.

## EXPERIMENTS AND RESULTS ANALYSIS

We evaluated our approach on six real networks and 25 artificial datasets, employing *EQ* (*Gao, 2023*) and *NMI* (*Wang et al., 2023*) metrics to compare dozens of classic and recent overlapping community discovery algorithms. *EQ* measures the quality of overlapping community detection by extending traditional modularity, while *NMI* quantifies the similarity between detected and ground-truth communities. We applied F1-score, Precision, and Recall (*Yang et al., 2023*) to assess performance on overlapping nodes and compared our results with five baseline algorithms. F1-score provides a balanced measure of precision and recall for overlapping node identification. We also analyzed the

**Table 1 Basic characteristics of real datasets.** The features of the real datasets are shown, where $|V|$ represents the number of nodes, $|E|$ denotes the number of edges, $|\Omega|$ is the number of real communities in the network, $max(k)$ is the maximum value of node degree, $<k>$ represents the average degree of nodes, $<d>$ denotes the average path length of the network, and $<c>$ signifies the average clustering coefficient of nodes.

| Data set | $|V|$ | $|E|$ | $|\Omega|$ | $max(k)$ | $<k>$ | $<d>$ | $<c>$ |
|---|---|---|---|---|---|---|---|
| Karate | 34 | 78 | 2 | 17 | 4.588 | 2.408 | 0.571 |
| Dolphin | 62 | 159 | 2 | 12 | 5.129 | 3.357 | 0.259 |
| Polbooks | 105 | 441 | 3 | 25 | 8.4 | 3.079 | 0.448 |
| Football | 115 | 613 | 12 | 12 | 10.661 | 2.508 | 0.403 |
| Power | 4,941 | 6,594 | — | 19 | 2.67 | 18.989 | 0.108 |
| Internet | 22,963 | 48,436 | — | 2,390 | 4.22 | 3.528 | 0.239 |

algorithm's convergence behavior, sensitivity to embedding dimensions and random walk parameters, and computational complexity to demonstrate its superiority.

## Experiment result analysis of real networks

### Real datasets

We used six real datasets in the experiment. These datasets come from the Stanford University datasets website. Table 1 shows the features of the datasets, where $|V|$ represents the number of nodes, $|E|$ denotes the number of edges, $|\Omega|$ is the number of real communities in the network, $max(k)$ is the maximum value of node degree, $<k>$ represents the average degree of nodes, $<d>$ denotes the average path length of the network, and $<c>$ signifies the average clustering coefficient of nodes.

### Comparison with classic overlapping community discovery algorithms

We compared and analyzed the algorithm proposed in this study with five classic overlapping community discovery algorithms on the six real datasets mentioned above. We ran each algorithm independently 100 times on the corresponding datasets and obtained the average *EQ* of the six algorithms across the six datasets, as shown in Fig. 1.

Experimental results demonstrate that GELPA-OCD consistently outperforms all five baseline algorithms across all datasets regarding *EQ* values. Especially on large-scale networks such as Power and the Internet, the performance of the proposed algorithm significantly improved, with an average improvement of 94.97% and 88.56%, respectively, demonstrating its superiority in handling large-scale networks. In the medium-scaled network, compared with other algorithms, the proposed algorithm improved *EQ* values by 4.77~38.31% (average improvement of 27.07%) on the Dolphin dataset, 2.66~18.55% (average improvement of 10.33%) on the Football dataset, and 3.02~46.05% (average improvement of 16.79%) on the Polbooks dataset. On the small-scale dataset Karate, the proposed algorithm achieved an average improvement of 39.68% in *EQ* values compared to the other five algorithms. The experimental results fully validated the proposed algorithm's effectiveness, correctness, and accuracy in discovering overlapping communities.

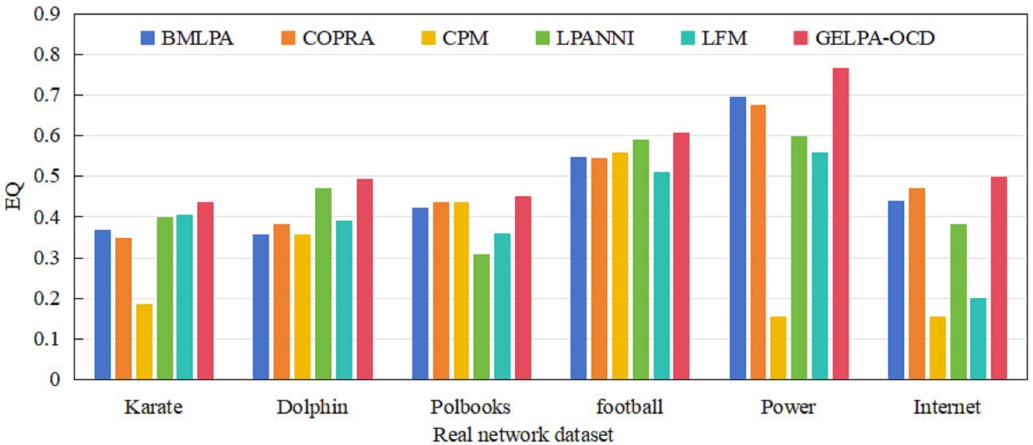

**Figure 1** **Comparison of overlapping modularity of six algorithms on six real datasets.** On the six real datasets mentioned above, the algorithm proposed in this article was compared and analyzed with five classic overlapping community discovery algorithms. Each algorithm was independently run 100 times on the corresponding datasets, and the average overlapping modularity of the six algorithms on the six datasets was obtained.

### Performance comparison with newly appearing algorithms

To further verify the correctness and effectiveness of the proposed algorithm for overlapping community detection, we conducted comparative analyses with seven recent community detection algorithms: CLEM (*Xue & Tang, 2020*), NIEM (*Yang et al., 2023*), EMOFM (*Tian, Yang & Zhang, 2019*), OCLN (*Yang et al., 2023*), CFCD (*Zhang, Ding & Yang, 2019*), CDSAT (*Jabbour et al., 2020*), and PCDSAT (*Jabbour et al., 2020*). We selected four classic datasets, Karate, Dolphin, Football, and Polbooks, and used the *EQ* value as the evaluation index. Table 2 shows the results.

Table 2 showed that the *EQ* values computed for the community discovery results of the eight algorithms on the four classic datasets were not completely the same. However, the GELPA-OCD algorithm proposed in this article performed well on all four classic networks, with *EQ* values higher than those of the other seven algorithms. The above results verified the proposed algorithm's effectiveness, correctness, and higher quality for overlapping community discovery.

### Algorithm stability analysis

We conducted 100 repeated experiments on the Karate and Polbooks datasets to evaluate algorithm stability, comparing GELPA-OCD with two classic algorithms, COPRA and LFM. Figure 2 presents the *EQ* values across all experiments.

As shown in Fig. 2, the proposed algorithm showed significant advantages in stabilizing community discovery results. On the Karate and Polbooks datasets, the GELPA-OCD algorithm's *EQ* values remained unchanged with minimal fluctuations, while the COPRA and LFM algorithms showed significant fluctuations. The consistency of EQ values across multiple runs is crucial as it indicates algorithmic stability and reliability. In practical applications, stable algorithms ensure that users obtain consistent community detection results regardless of random initialization, which is essential for reproducible research and

**Table 2 EQ values of community discovery results of eight algorithms in classical networks.** The algorithm GELPA-OCD proposed in this paper is compared with seven recent community discovery algorithms. Four classic datasets, karate, dolphin, football and polbooks, are selected as examples, and the overlapping modularity EQ is used as the evaluation index.

| Algorithms | EQ | | | |
|---|---|---|---|---|
| | Karate | Dolphin | Polbooks | Football |
| CLEM | 0.361 | 0.3568 | 0.338 | 0.574 |
| NIEM | 0.3505 | 0.4366 | 0.4423 | 0.5302 |
| EMOFM | 0.2341 | 0.2723 | 0.2703 | 0.3066 |
| OCLN | 0.3584 | 0.3332 | 0.4607 | 0.3687 |
| CFCD | 0.3718 | 0.4938 | 0.4426 | 0.6005 |
| CDSAT | 0.311 | 0.297 | 0.345 | 0.404 |
| PCDSAT | 0.396 | 0.324 | 0.345 | 0.420 |
| GELPA-OCD | 0.4498 | 0.4999 | 0.4698 | 0.6028 |

**Table 3 Basic characteristics of artificial datasets.**

| Group | Network name | $|V|$ | $k$ | $\mu$ | $max(k)$ | $min|\Omega|$ | $max|\Omega|$ | $|on|$ | $|om|$ |
|---|---|---|---|---|---|---|---|---|---|
| A | LFR1~LFR5 | 1,000 | 10 | 0.1~0.5 | 50 | 10 | 50 | 100 | 2 |
| B | LFR6~LFR12 | 1,000 | 10 | 0.1 | 50 | 10 | 50 | 200~500 | 2 |
| C | LFR13~LFR18 | 1,000 | 10 | 0.1 | 50 | 10 | 50 | 100 | 3~8 |
| D | LFR19~LFR25 | 2,000~5,000 | 10 | 0.1 | 50 | 10 | 50 | 100 | 2 |

**Table 4 Comparison of *Precision, Reacll, and F1-score* among different algorithms.**

| Network name | $|V|$ | Metric(%) | COPRA | BMLPA | CPM | LPANNI | LFM | GELPA-OCD |
|---|---|---|---|---|---|---|---|---|
| LFR19 | 2,000 | *Precision* | 96.6 | 100 | 44.1 | 98.7 | 12.4 | 98.9 |
| | | *Recall* | 85.3 | 45.5 | 35.4 | 74.6 | 45.2 | 88.0 |
| | | *F1-score* | 90.59 | 65.54 | 39.25 | 84.97 | 19.46 | 93.15 |
| LFR20 | 2,500 | *Precision* | 97.7 | 96.4 | 35.8 | 98.3 | 25.6 | 98.8 |
| | | *Recall* | 86.5 | 53.5 | 45.8 | 59.0 | 25.5 | 89.5 |
| | | *F1-score* | 91.76 | 68.80 | 40.19 | 73.74 | 25.55 | 93.91 |
| LFR21 | 3,000 | *Precision* | 96.8 | 94.9 | 45.8 | 94.4 | 24 | 98.9 |
| | | *Recall* | 83.1 | 38.1 | 31.3 | 67.3 | 32.5 | 89.5 |
| | | *F1-score* | 89.41 | 54.36 | 37.17 | 78.59 | 27.61 | 93.93 |
| LFR22 | 3,500 | *Precision* | 85.7 | 85.8 | 35.6 | 93.1 | 36 | 89.6 |
| | | *Recall* | 78.7 | 50.0 | 46.9 | 67.3 | 30.5 | 86.7 |
| | | *F1-score* | 82.05 | 63.18 | 40.48 | 78.10 | 33.01 | 88.13 |
| LFR23 | 4,000 | *Precision* | 88.1 | 82.3 | 35.4 | 97.1 | 0 | 100 |
| | | *Recall* | 82.0 | 54.2 | 40.1 | 60.9 | 26.5 | 89.5 |
| | | *F1-score* | 84.95 | 65.35 | 37.58 | 74.84 | 0 | 94.46 |
| LFR24 | 4,500 | *Precision* | 97.8 | 77.6 | 0 | 93.2 | 15 | 100 |
| | | *Recall* | 89.5 | 50.3 | 0 | 59.6 | 21.2 | 90.4 |
| | | *F1-score* | 94.45 | 61.02 | 0 | 72.69 | 17.57 | 94.96 |

| Network name | \|V\| | Metric(%) | COPRA | BMLPA | CPM | LPANNI | LFM | GELPA-OCD |
|---|---|---|---|---|---|---|---|---|
| LFR25 | 5,000 | *Precision* | 100 | 70.7 | 54.2 | 100 | 3 | 95.6 |
| | | *Recall* | 89.1 | 61.0 | 32.1 | 61.8 | 18.1 | 89.2 |
| | | *F1-score* | 94.23 | 65.47 | 40.32 | 76.39 | 5.15 | 92.29 |

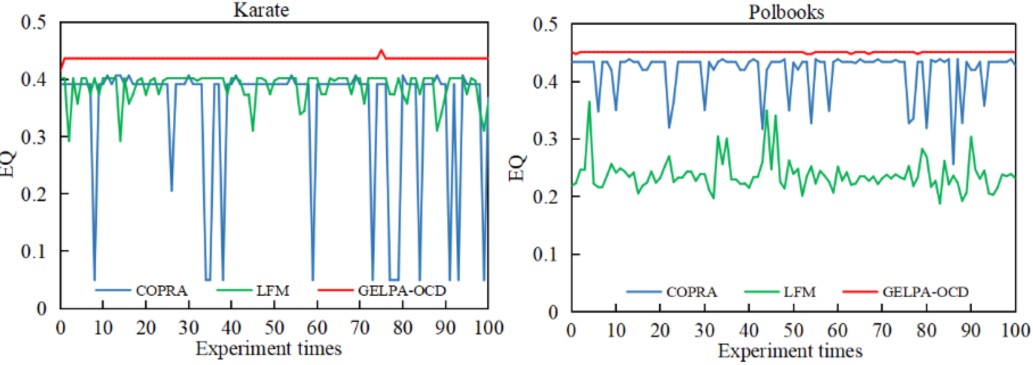

**Figure 2 Comparison of modularity of community partitioning results among three algorithms.** To evaluate algorithm stability, we conducted 100 repeated experiments on the Karate and Polbooks datasets, comparing GELPA-OCD with two classic algorithms, COPRA and LFM.

reliable decision-making processes. The experimental results showed that the proposed algorithm effectively improved the randomness of node update order by introducing a multidimensional feature-based node importance evaluation index. The improved label propagation process used similarity weights and adaptive label filtering thresholds, effectively improving algorithm stability and enhancing the quality of community discovery.

### Analysis of algorithm convergence

To verify the convergence of the proposed algorithm, we recorded label changes of network nodes with increasing iteration numbers for both the proposed GELPA-OCD algorithm and comparative algorithms. The maximum number of iterations in the experiments was set to 30 for small and 300 for large datasets. Taking the classic Karate and Football datasets as examples, Fig. 3 illustrates the convergence process of three algorithms, with the horizontal axis representing the number of iterations and the vertical axis showing the proportion of nodes whose labels changed in the current iteration relative to the total number of nodes in the network, *i.e.*, the label change rate.

As shown in Fig. 3, all algorithms converged within 10 iterations, but GELPA-OCD converged faster. On the Karate network, convergence reached just four iterations, compared to 6 and 10 for BMLPA and COPRA, respectively. On the more complex Football network, GELPA-OCD achieved near-zero label change by the 3rd iteration, while BMLPA and COPRA required 7 to 10 iterations. These results confirmed the proposed algorithm's faster convergence and improved efficiency.

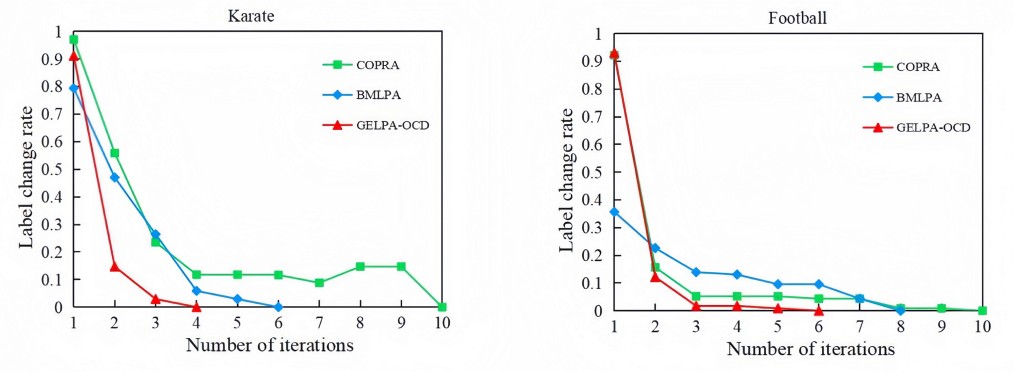

**Figure 3 Convergence comparison of three algorithms.**

## Analysis of experimental results of artificial datasets
### Artificial datasets
Due to the unclear community structure of many real network datasets, it is not easy to effectively measure the accuracy and effectiveness of algorithms. Therefore, we used an LFR benchmark (*Lancichinetti, Fortunato & Radicchi, 2008*) to generate four groups of artificial networks: A, B, C, and D. The total number of artificial datasets is 25. Table 3 shows the basic information on artificial networks. Among them, $|V|$ is the number of network nodes, and the algorithm in this article takes values from 1,000 to 5,000; The mixing parameter $\mu$ is defined as $\mu = \dfrac{d_{out(v)}}{d_v}$, which measures the ratio of external to total connections for each node, with values ranging from 0.1 to 0.5 in this article; $k$ is the average degree of nodes, with a uniform value of 10; $max(k)$ is the maximum degree of the network, with a uniform value of 50; $min|\Omega|$ represents the minimum size of the community, with a uniform value of 10; $max|\Omega|$ represents the maximum size of the community, with a uniform value of 50; $|on|$ is the number of overlapping nodes in the network, with values ranging from 100 to 500 in this article; $|om|$ represents the number of communities to which overlapping nodes belong, with values ranging from 2 to 8 in this article.

### Comparison of NMI values for community discovery of different algorithms
We compared the proposed algorithm with five others on four groups of synthetic datasets, using $NMI$ as the evaluation metric to assess performance across different network structures. Figure 4 shows the results.

Figure 4 shows that the proposed algorithm performed best on the four artificial networks, followed by the LPANNI algorithm. In Group A, as the mixing parameter increased, all algorithms showed declining $NMI$ values, but our algorithm maintained the highest scores throughout, demonstrating strong robustness in fuzzy community structures. In Group B, increasing the number of overlapping nodes from 200 to 500 made the networks more complex. Although all methods showed reduced performance, our algorithm consistently achieved the highest $NMI$. In Group C, as the number of

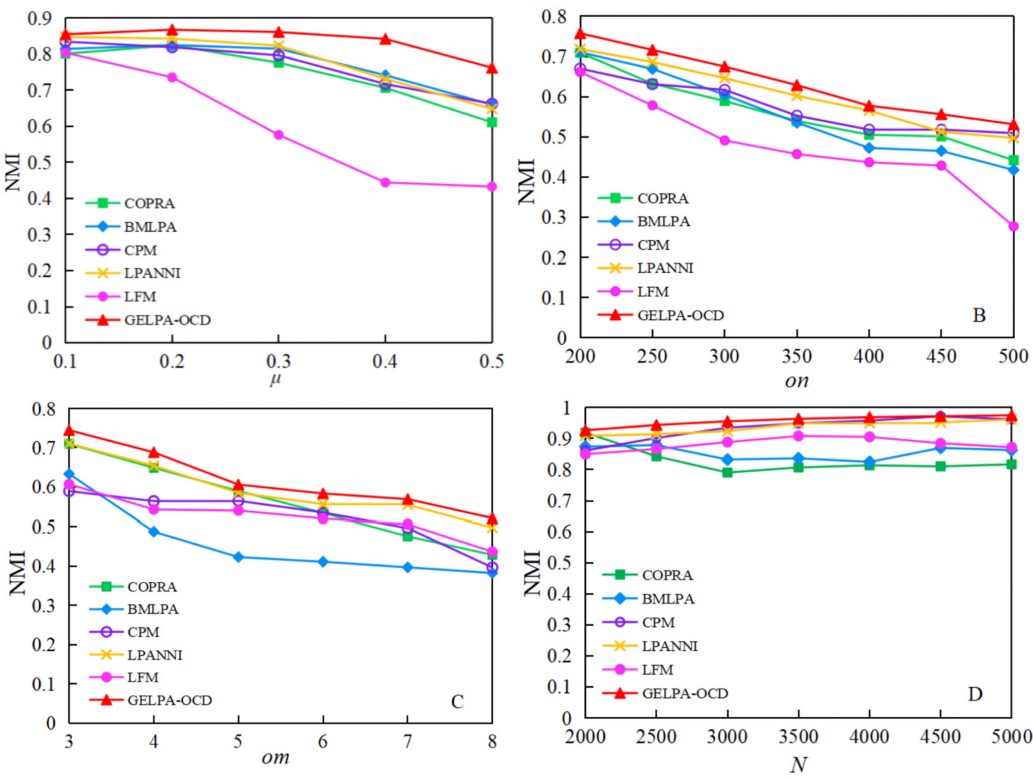

**Figure 4 Comparison of *NMI* of six algorithms on artificial datasets.** To verify the performance of the algorithm in networks with different structural features, a comparative experiment was conducted among the algorithm proposed in this article and five other algorithms on four groups of artificial datasets, with *NMI* as the evaluation index.         

communities per overlapping node increased, all algorithms experienced performance drops, with BMLPA and CPM declining the most. Our algorithm showed the slowest decline, indicating superior performance in highly overlapping scenarios. In Group D, as the network grew from 2,000 to 5,000 nodes, the proposed algorithm's *NMI* values remained stable and slightly increased, confirming its scalability and strong performance on large-scale networks.

### Quality assessment of overlapping node detection

To more comprehensively evaluate the effectiveness and accuracy of the proposed algorithm in identifying overlapping nodes, precision, recall, and F1-score were used as evaluation metrics to compare the overlapping node identification performance of different algorithms on seven artificially synthesized networks of group D. This group of networks maintained the same mixing parameter ($\mu = 0.1$) and number of overlapping nodes ($on = 100$), while varying the network scale. Table 4 compares precision, recall, and F1-score for overlapping node identification by six algorithms on seven artificial networks of different scales.

As shown in Table 4, GELPA-OCD consistently achieved excellent performance. It maintained high precision across all networks, while other algorithms showed fluctuating

results. For instance, BMLPA reached 100% precision on the LFR19 network but dropped sharply on larger networks. CPM and LFM performed poorly, with precision falling below 50% and even down to 0 on some networks. Regarding recall, GELPA-OCD stayed above 86% across all networks, with an average of 89%, indicating strong coverage. COPRA followed with an average recall of 84.9%. In contrast, LPANNI showed inconsistent results, and CPM and LFM performed poorly, especially with CPM achieving 0% recall on LFR24 and LFM falling under 30% on several large networks. For the F1-score, which balances precision and recall, GELPA-OCD again led, averaging 93% and showing strong stability across different scales. COPRA ranked second, while LPANNI was stable but underperformed compared to GELPA-OCD. CPM and LFM showed significant instability, with the F1-score dropping to 0 on some networks. Overall, GELPA-OCD demonstrated superior performance in community detection and overlapping node identification, particularly in complex and large-scale network environments.

## Node importance evaluation and model performance analysis

### Ablation experiments on node importance metric combination strategies

To verify the effectiveness of this article's node importance metric combination method, we compared the community detection performance of four combination strategies: multiplicative combination, additive combination, weighted combination, and nonlinear combination of the proposed node importance metrics, under identical conditions. Formulas (5), (11), (12), and (13) show the multiplicative combination (the combination method used in this article), additive combination, weighted combination, and nonlinear combination, respectively.

$$NI(v) = DC\_norm(v) + PR\_norm(v) + (1 + CC\_norm(v)) \tag{11}$$
$$NI(v) = \omega_1 * DC\_norm(v) + \omega_2 * PR\_norm(v) + \omega_3(1 + CC\_norm(v)) \tag{12}$$
$$NI(v) = DC\_norm(v) + PR\_norm(v)^{\alpha} + (1 + CC\_norm(v))^{\beta} \tag{13}$$

where $NI(v)$ represents the importance of node $v$. The calculation methods for $DC\_norm(v)$, $PR\_norm(v)$, and $CC\_norm(v)$ are shown in Formulas (6), (7), and (8) above. $\omega_1$, $\omega_2$ and $\omega_3$ are weight coefficients, where $\omega_1 + \omega_2 + \omega_3 = 1$. $\alpha$ and $\beta$ are weight coefficients for the nonlinear combination, where $\alpha + \beta = 1$, facilitating normalized comparison. To ensure the effectiveness of the node importance metric combination method proposed in this article, we conducted a parameter sensitivity analysis for weighted and nonlinear combinations in the experiments, using the Karate and Dolphin networks as examples. We ran each set of weights independently 10 times and recorded the $EQ$ values. Table 5 shows the $EQ$ values for different weights.

As shown in Table 5, different weighting strategies slightly affected algorithm performance. We selected the parameter combination with the highest $EQ$ value, (0.3, 0.3, 0.4) for the weighted method and (0.3, 0.7) for the nonlinear method.

To verify the effectiveness and superiority of the multiplicative combination method of node importance metrics proposed in this algorithm, we conducted experiments on two classic datasets, Karate and Dolphin. We maintained other parameters of the proposed algorithm unchanged while only altering the calculation method for node importance.

**Table 5  EQ values of the algorithm under different weight combinations.**

| $(\omega_1,\ \omega_2,\ \omega_2)$ | EQ | | $(\alpha,\ \beta)$ | EQ | |
|---|---|---|---|---|---|
| | Karate | Dolphin | | Karate | Dolphin |
| (0.4, 0.4, 0.2) | 0.3916 | 0.3741 | (0.5, 0.5) | 0.4259 | 0.3785 |
| (0.3, 0.2, 0.5) | 0.3990 | 0.3957 | (0.4, 0.6) | 0.4366 | 0.3957 |
| (0.5, 0.3, 0.2) | 0.4070 | 0.4082 | (0.3, 0.7) | 0.4155 | 0.4168 |
| (0.3, 0.3, 0.4) | 0.4173 | 0.4740 | (0.7, 0.3) | 0.3990 | 0.3192 |
| (0.6, 0.2, 0.2) | 0.4165 | 0.3594 | (0.6, 0.4) | 0.4070 | 0.3741 |
| (0.2, 0.6, 0.2) | 0.4022 | 0.3748 | (0.8, 0.2) | 0.3155 | 0.4900 |

Each combination strategy was run independently 10 times, with averages taken to reduce the impact of randomness.

The nonlinear, additive, weighted, and multiplicative combination strategies for the karate dataset achieved $EQ$ values of 0.4071, 0.4242, 0.4032, and 0.4498, respectively. The $EQ$ values of the four combination strategies on the dolphin dataset were 0.4065, 0.3999, 0.4461, and 0.4962, respectively. The multiplication combination strategy achieved the highest $EQ$ value on the two classical datasets. The multiplication combination strategy selected in this article avoids the subjectivity of artificial parameter adjustment. Although the multiplication combination is sensitive to a single indicator near zero, the value range of the three basic indicators is controlled between [0, 1] after normalization. There are a few extreme cases near zero. In addition, the form of $(1 + CC\_norm(v))$ is adopted to ensure that even if the clustering coefficient is zero, the overall importance of the node will not be zero. The above experiments proved the superiority of the node importance index under the multiplication combination strategy proposed in this article.

### Parameter sensitivity analysis of the graph embedding algorithm

To investigate the impact of key parameters in Node2vec on the performance of the proposed algorithm, we conducted experiments on a small-scale dataset, Karate, a medium-scale dataset, Football, and large-scale datasets, Power and Internet, using $EQ$ as the evaluation metric, with a focus on analyzing the effects of embedding dimension and random walk parameters $p$ and $q$ on algorithm performance.

(1) Impact of embedding dimension on algorithm performance. To investigate the impact of embedding dimension on algorithm performance, we fixed random walk parameters at $p = 1.0$ and $q = 1.5$ and tested five different embedding dimensions: 32, 64, 128, 256, and 512. As shown in Fig. 5, the horizontal axis $D$ represents the embedding dimension, while the vertical axis represents the $EQ$ values corresponding to the algorithm's community detection results under different embedding dimensions.

As shown in Fig. 5, increasing the embedding dimension generally improved community detection quality up to a point, after which performance stabilized or declined. On the Karate network, the $EQ$ value increased from 0.417 to 0.45 as the dimension rose from 32 to 64, but declined beyond 128, suggesting that excessively high dimensions may cause overfitting. On the Internet network, $EQ$ values kept improving but with diminishing

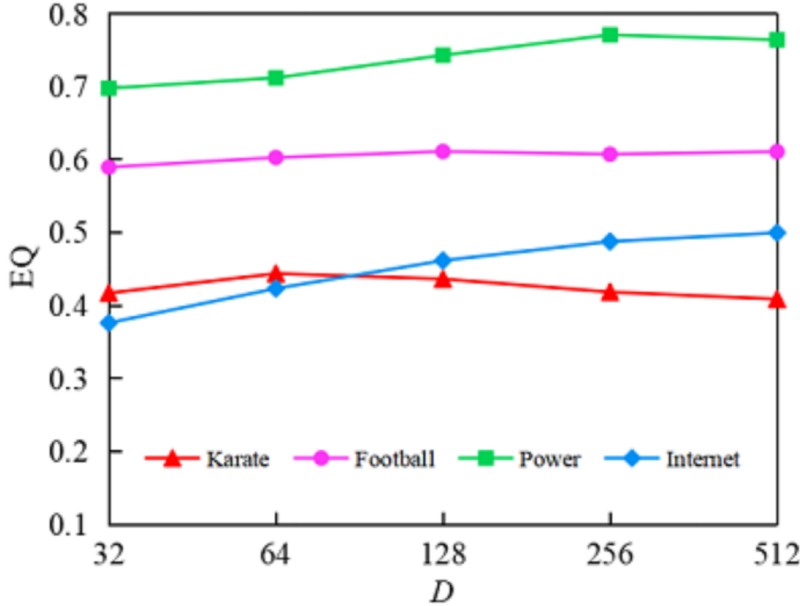

**Figure 5 Comparison of *EQ* values for community detection results of the GELPA-OCD in diûerent dimensions.** The effect of embedding dimension on algorithm performance. The results show that optimal dimensions vary by betwork scale: 64 for small networks (<100 nodes), 128 for medium networks (100–1,000 nodes), and 256 for large networks (>1,000 nodes). Performance generally improves with dimension up to a threshold, beyond which overfitting may occur.

gains beyond 256. The Power and Football networks showed more stable trends, with optimal performance typically at 128 and 256 dimensions. Based on the experiments in this experiment of this article, for small networks (nodes < 100), we can achieve relatively good results by selecting a dimension of 64; for medium-scale networks (nodes between 100–1,000), we typically choose a dimension of 128, while for large-scale networks (nodes > 1,000), selecting a dimension of 256 achieves a better balance between performance and computational efficiency.

(2) Impact of random walk parameters on algorithm performance. We conducted experiments on three representative networks, Karate, Football, and Power, to evaluate how Node2vec's random walk parameters $p$ and $q$ affected community detection. Parameter $p$ controls the probability of returning to the previous node, while $q$ influences the exploration of new nodes. In our experiment, we fixed the embedding dimension at 128 and varied $p$ and $q$ to observe their impact. Table 6 shows the experimental results.

From the experimental results shown in Table 6, we observed that random walk parameters had a certain impact on community detection quality. On the Football network, when $p = 1.0$ and $q = 1.5$, the algorithm achieved the optimal *EQ* value of 0.616. This parameter combination leveraged the ability to capture structural equivalence when $q < 1$, which was beneficial for identifying relatively distinct community boundaries in the network. For the Karate network, the algorithm performed best ($EQ = 0.449$) when $p = 1.0$ and $q = 1.5$, indicating that small-scale networks required appropriately enhanced

**Table 6 Impact of Node2vec random walk parameters (p, q) on community detection quality: *EQ* values across three representative networks (Karate, Football, Power) with fixed embedding dimension *D* = 128.** To investigate the influence of random walk parameters *p* and *q* in the Node2-vec model on community detection quality, comparative experiments were conducted on three representative networks: Karate, Football, and Power.

| Parameters | | EQ | | |
| --- | --- | --- | --- | --- |
| *p* | q | Karate | Football | Power |
| 0.5 | 0.5 | 0.417 | 0.582 | 0.731 |
| 0.5 | 1.0 | 0.425 | 0.590 | 0.742 |
| 0.5 | 1.5 | 0.418 | 0.583 | 0.735 |
| 0.5 | 2.0 | 0.409 | 0.578 | 0.729 |
| 1.0 | 0.5 | 0.436 | 0.616 | 0.756 |
| 1.0 | 1.0 | 0.438 | 0.608 | 0.762 |
| 1.0 | 1.5 | 0.449 | 0.602 | 0.771 |
| 1.0 | 2.0 | 0.425 | 0.595 | 0.742 |
| 1.5 | 0.5 | 0.428 | 0.601 | 0.748 |
| 1.5 | 1.0 | 0.430 | 0.597 | 0.745 |
| 1.5 | 1.5 | 0.422 | 0.589 | 0.739 |
| 1.5 | 2.0 | 0.416 | 0.581 | 0.732 |
| 2.0 | 0.5 | 0.408 | 0.593 | 0.740 |
| 2.0 | 1.0 | 0.411 | 0.586 | 0.736 |
| 2.0 | 1.5 | 0.405 | 0.580 | 0.730 |
| 2.0 | 2.0 | 0.407 | 0.575 | 0.723 |

stability in label propagation to avoid excessive focus on local structures. For the medium-scale Power network, the parameter combination of $p = 1.0$ and $q = 1.0$ showed optimal performance ($EQ = 0.762$). This balanced random walk strategy was more suitable for handling larger and structurally complex networks. When the $p$ and $q$ parameters deviated from the optimal combination, algorithm performance varied across all three networks.

Overall, GELPA-OCD performed consistently well when $p = 1.0$ and q ranged between 0.5 and 1.5. These findings confirmed that Node2vec's biased walks significantly influenced overlapping community detection. Accordingly, our main experiments used $p = 1.0$ and $q = 1.5$.

### Performance comparison of different graph embedding algorithms

To validate the choice of Node2vec as the graph embedding method, we compared its performance with the classic Deepwalk algorithm on four benchmark datasets. We kept embedding dimensions, walk length, and number of walks consistent across both methods and varied only the $p$ and $q$ parameters for Node2vec. Table 7 presents the *EQ* values and the number of detected communities.

The results showed that Node2vec consistently outperformed Deepwalk, achieving higher *EQ* values across all datasets. On the Karate and Dolphin networks, our algorithm detected three communities instead of the actual two, but achieved higher modularity

**Table 7 Comparison of experimental results of two graph embedding algorithms on four datasets.** In the experiments in this paper, based on two different graph embedding algorithms, comparisons of community detection quality were conducted on four classic datasets, maintaining consistent parameter settings for embedding dimensions, walk length, and number of walks, differentiating the two methods only by adjusting Node2vec's $p$ and $q$ parameters.

| Network | Karate | | Dolphin | | Polbooks | | Football | |
|---|---|---|---|---|---|---|---|---|
| Criteria | $|\Omega|$ | EQ | $|\Omega|$ | EQ | $|\Omega|$ | EQ | $|\Omega|$ | EQ |
| DeepWalk | 5 | 0.4158 | 2 | 0.3957 | 3 | 0.4439 | 8 | 0.6057 |
| Node2Vec | 3 | 0.4498 | 3 | 0.4963 | 3 | 0.4695 | 10 | 0.6158 |

while maintaining stability and efficiency. On the Football network, the Node2vec-based method produced results closest to the actual number of communities. These findings validate Node2vec as an appropriate embedding technique, with its parameterized random walks effectively balancing node homogeneity and structural equivalence, providing more accurate node similarity information for label propagation.

### Theoretical analysis of adaptive threshold filtering mechanism

The adaptive threshold $\theta = \frac{1}{|L(v)|}$ proposed in this article improved the label propagation mechanism. From the perspective of algorithm convergence, this threshold ensured monotonicity in the label propagation process; in each iteration, labels with affiliation coefficients below the "average level" were filtered out, causing the retained label affiliation coefficients to become more concentrated on dominant communities after normalization, thereby accelerating the algorithm's convergence speed. In this article's experimental section, "Analysis of algorithm convergence," convergence analysis was conducted on the classic Karate and Football datasets, comparing the proposed algorithm with three classic overlapping community detection algorithms. Compared to COPRA and BMLPA algorithms, the proposed algorithm demonstrated faster label distribution change rate convergence, typically reaching a stable state after 3 or 4 iterations.

Furthermore, this threshold mechanism could automatically balance the precision and recall of label retention in networks with different topological structural properties. In densely connected areas (within communities), nodes typically received multiple labels (larger $|L(v)|$), resulting in a lower threshold that allowed nodes to retain multiple labels with higher affiliation coefficients, ensuring high recall. In sparsely connected areas (community boundaries), nodes received fewer labels (smaller $|L(v)|$), resulting in a higher threshold that filtered out most labels with low affiliation coefficients through this mechanism, improving the precision of label selection. In this article's experimental section, "Assessment of overlapping node detection," experimental results on artificial datasets of different scales showed that the proposed algorithm demonstrated good recall and precision in identifying overlapping nodes.

## ALGORITHM COMPLEXITY ANALYSIS

To analyze the complexity of the proposed algorithm, $n$ represents the total number of nodes in the network, $m$ represents the number of edges, $d$ represents the average degree of

**Table 8 Runtime performance comparison of GELPA-OCD and baseline algorithms (seconds).**

| Dataset | BMLPA | COPRA | CPM | LPANNI | LFM | GELPA-OCD |
|---|---|---|---|---|---|---|
| Karate | 0.57 | 0.53 | 3.51 | 0.75 | 1.58 | 1.45 |
| Dopphin | 1.35 | 1.25 | 9.80 | 1.55 | 2.95 | 2.85 |
| Polbooks | 1.51 | 1.48 | 25.32 | 1.87 | 13.33 | 8.97 |
| Football | 3.57 | 3.15 | 180.45 | 5.45 | 120.11 | 35.14 |
| Power | 135.30 | 75.25 | 7,200 | 300.75 | 1,400 | 750 |
| Internet | 600.35 | 552.07 | 33,285 | 800.31 | 8,935 | 1,800 |

nodes, $D$ represents the dimension of embedding vectors, $t$ represents the maximum number of algorithm iterations, $l$ represents the length of random walks, $w$ represents the number of random walks, $r$ represents the overlap degree, and $k$ represents the size of cliques in the CPM.

First, in the node importance calculation phase, the proposed algorithm needs to calculate degree centrality, PageRank value, and clustering coefficient, with computational complexities of $O(m)$, $O(t \cdot (n + m))$, and $O(n \cdot d^2)$, respectively. Second, using Node2vec for graph embedding, the time complexity is $O(n \cdot w \cdot l \cdot D)$. Third, in the label propagation phase, calculating the similarity matrix between nodes requires a complexity of $O(n^2 \cdot D)$. Finally, label iteration has a time complexity of $O(t \cdot n \cdot d)$. Therefore, the total time complexity of the algorithm in this article is $O(t \cdot (n + m) + n \cdot w \cdot l \cdot D + n^2 \cdot D)$. The complexity of the COPRA algorithm is $O(r \cdot t \cdot n \cdot d)$. The time complexity of the CPM algorithm can reach an exponential level in the worst case. The complexity of the LFM algorithm is $O(n^2 log(n))$, showing lower efficiency when processing large networks. The time complexity of the LPANNI algorithm is $O(t \cdot n \cdot d)$, but it performs poorly in networks with fuzzy community structures. The BMLPA algorithm effectively reduced the randomness of the algorithm, with a time complexity of $O(t \cdot n \cdot d)$. The GELPA-OCD algorithm proposed in this article has a theoretical complexity of $O(t \cdot (n + m) + n \cdot w \cdot l \cdot D + n^2 \cdot D)$. Although it introduces node importance calculation and graph embedding steps, making the overall computational complexity higher than algorithms such as LPANNI, COPRA, and BMLPA, it addresses the issues of randomness in node update order and uncertainty in label selection in traditional LPA-based algorithms, significantly improving the quality and stability of community partitioning.

## RUNTIME PERFORMANCE COMPARISON

To evaluate the practical computational efficiency of the proposed algorithm, we conducted runtime experiments on representative datasets and compared GELPA-OCD with five baseline algorithms (BMLPA, COPRA, CPM, LPANNI, and LFM). Table 8 presents the runtime (in seconds).

As shown in Table 8, GELPA-OCD demonstrates competitive runtime on small to medium-scale networks. Although it requires higher computational time on

large-scale datasets (Power and Internet), this overhead is justified by significant performance improvements. As shown in Fig. 1, GELPA-OCD achieves substantial EQ value enhancements on these large networks, with average improvements of 94.97% on Power and 88.56% on Internet datasets compared to baseline algorithms. This represents a favorable trade-off between computational complexity and community detection quality for applications prioritizing accuracy over processing speed.

## CONCLUSIONS

This article presents the GELPA-OCD algorithm that integrates graph embedding and label propagation. The algorithm adopts a fixed node update strategy, proposes a new method for measuring node importance, and develops a weighted community membership coefficient formula. Experimental validation on diverse real-world and synthetic datasets demonstrates significant improvements in stability and accuracy compared to existing methods. The algorithm effectively handles networks with overlapping community structures, achieving significant improvements in detection accuracy and algorithmic stability.

This study focuses on undirected and unweighted networks. We plan to extend the proposed algorithm to directed and weighted graphs in future work. We will modify the Node2vec random walk strategy to consider edge weight and direction information. While our evaluation encompasses diverse real-world and synthetic datasets, overlapping communities exhibit domain-specific semantic characteristics that warrant further investigation. Future research will continue to explore the impact of the algorithm in specific application domains.

## ACKNOWLEDGEMENTS

We thank the editors for taking the time to edit this article, and would also like to thank the reviewers for their constructive comments, which were very helpful in strengthening the presentation of this study.

### Funding

This work was supported by the National Natural Science Foundation of China (No. 42002138, 62172352, 62171143), the Natural Science Foundation of HeBei Province (No. D2023107002), the Hebei Province Central Leading Local Science and Technology Development Project (No. 246Z1817G), the Special Project of Northeast Petroleum University Characteristic Domain Team (No. 2022TSTD-03), the Postdoctoral Scientific Research Development Fund of Heilongjiang Province (No. LBH-Q20073), and the Basic Research Support Plan for Excellent Young Teachers in Heilongjiang Province (No. YQJH2023073). The funders had no role in study design, data collection and analysis, decision to publish, or preparation of the manuscript.

## Grant Disclosures

The following grant information was disclosed by the authors:

National Natural Science Foundation of China: 42002138, 62172352, 62171143.

Natural Science Foundation of HeBei Province: D2023107002.

Hebei Province Central Leading Local Science and Technology Development Project: 246Z1817G.

Special Project of Northeast Petroleum University Characteristic Domain Team: 2022TSTD-03.

Postdoctoral Scientific Research Development Fund of Heilongjiang Province: LBH-Q20073.

Basic Research Support Plan for Excellent Young Teachers in Heilongjiang Province: YQJH2023073.

## Competing Interests

The authors declare that they have no competing interests.

## Author Contributions

- Miaomiao Liu conceived and designed the experiments, authored or reviewed drafts of the article, and approved the final draft.
- Yuchen Liu conceived and designed the experiments, performed the experiments, performed the computation work, prepared figures and/or tables, authored or reviewed drafts of the article, and approved the final draft.
- Yanan Hu analyzed the data, prepared figures and/or tables, and approved the final draft.
- Jing Chen performed the computation work, prepared figures and/or tables, and approved the final draft.
- Wenqing Zhang analyzed the data, prepared figures and/or tables, and approved the final draft.

## Data Availability

The data and code are available in the Supplemental File.

## Supplemental Information

Supplemental information for this article can be found online at http://dx.doi.org/10.7717/peerj-cs.3389#supplemental-information.

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
