# Peer review of "Overlapping community discovery based on graph embedding and label propagation algorithm"

_PeerJ Computer Science, doi:10.7717/peerj-cs.3389_

## Round 0.1 · original submission · Major Revisions

· Academic Editor

Major Revisions

**Language Note:** The review process has identified that the English language must be improved. PeerJ can provide language editing services - please contact us at [email protected] for pricing (be sure to provide your manuscript number and title). Alternatively, you should make your own arrangements to improve the language quality and provide details in your response letter. – PeerJ Staff

·

Basic reporting

This paper introduces a novel approach to overlapping community discovery by integrating graph embedding and a label propagation algorithm, termed GELPA-OCD. The manuscript contributes meaningfully to the literature by addressing persistent challenges in traditional label propagation-based methods, such as instability due to random node selection and label updates, as well as inadequate modeling of node importance and similarity. By proposing a new multi-factor node importance metric (combining degree centrality, PageRank, and clustering coefficient), leveraging Node2vec for graph embedding to capture deeper structural information, and introducing a similarity-weighted label attribution strategy with adaptive filtering, the method enhances both the accuracy and stability of community detection. The authors validate their approach using extensive experiments on six real-world datasets and 25 synthetic networks, demonstrating significant improvements in EQ and NMI over multiple baselines.

a. The formulation of the node importance index is a major strength, yet its multiplication-based design may suffer from sensitivity to any individual normalized metric approaching zero. A comparative ablation study evaluating additive, weighted, or nonlinear formulations would enhance the theoretical justification of this choice.

b. The graph embedding component relies solely on Node2vec. While this is justified by its flexibility, the exclusion of comparative results using other state-of-the-art embeddings such as DeepWalk or GraphSAGE weakens the generalizability claim. Including such baselines could strengthen the methodological rigor.

c. The adaptive threshold filtering mechanism lacks a formal analysis of its convergence behavior and stability guarantees. It would benefit the reader to understand how the threshold dynamically balances precision and recall of label retention across diverse graph topologies.

d. Although the algorithm improves label attribution using cosine similarity, the impact of noise and embedding dimensionality on this similarity measure is not sufficiently explored. A sensitivity analysis or variance study on the embedding parameters (p, q, dimensions) would provide practical guidance.

e. The experiments report strong EQ and NMI improvements, but the manuscript does not present time complexity or runtime comparisons with competing algorithms. Given that PageRank and embedding computations are resource-intensive, a scalability evaluation on larger networks is necessary for real-world applicability.

f. The current design assumes unweighted and undirected graphs. However, many real-world networks contain edge weights and directions. Discussing how the proposed method could be extended to accommodate these cases would broaden its potential impact.

Experimental design

g. The use of Node2vec’s BFS/DFS biased walk parameters is briefly mentioned but not elaborated on in the context of overlapping detection. A more focused discussion on how varying walk strategies influence community boundaries in embeddings would improve interpretability.

h. While the algorithm is benchmarked across several datasets, the paper could further distinguish itself by exploring application-specific implications—such as in biological or citation networks—where overlapping community semantics differ. This would help contextualize the algorithm's relevance beyond structural metrics.

The Literature citation is not adequate, and the related work to machine learning should be discussed
1.Sdac-da: Semi-supervised deep attributed clustering using dual autoencoder
2.A Comprehensive Survey on Spectral Clustering with Graph Structure Learning
3.Sparse feature selection using hypergraph Laplacian-based semi-supervised discriminant analysis

Validity of the findings

f. The current design assumes unweighted and undirected graphs. However, many real-world networks contain edge weights and directions. Discussing how the proposed method could be extended to accommodate these cases would broaden its potential impact.

·

Basic reporting

There are typos and errors in the paper that need to be corrected as detailed below. Otherwise, all aspects of basic reporting - the English, background of the paper, figures, tables, data shared, results, and definitions are acceptable, but would benefit from incorporating the suggestions provided below.

- English language of the paper can be improved, e.g. grammar of line 89 and 460 is off. There are typos, e.g. in line 272 - it should be combining instead of combing, update_dequence in line 559 is wrong, in algorithm 2 the word normalization is misspelled, and in the supplemental folder, the folder name should be 03Real_Datasets_Output. Clarity in writing can also be improved, e.g. line 537 is not clear. Line 551 and the lines after do not fit in the section they are in - it would be better to have them in a different summary section. The language in the paper, esp. relating to the conclusions is repetitive, and can be condensed to sound more professional.
- References have some inconsistencies - some have [C] (e.g. ref 15) and [J] (e.g. ref 4) but others don't. Minor comment - Since the full references have been spelled out in the text, adding brackets before and after them would help differentiate them from the main text to help make reading easier for reviewers.
- Table 4 has the text "Error! Reference source not found" for all the algorithms and needs to be fixed.
- Table and figure legends can be improved with more details, e.g. defining the letters k, d, c used in the table header in Table 3. A general tip is to make figures and tables as self-explanatory as possible without having to refer the main text, since many people look at the figures and tables first before reading the paper in detail.
- Table 1 algorithm title (label 'update') is inconsistent with the legend (label 'initialization') which can be confusing since figure 2 legend is also named 'label update'.
- Figure 1, figure 8, table 5 legends have spaces between letters in words.
- Equation 12 can be improved - The letter l should be described, answering which labels are covered. Line 539 talks about the label set but does not define it there - consider moving equation 12 up to this line.

Experimental design

The experimental design of the paper is satisfactory.

The paper's research is within the scope of the journal and addresses the relevant and meaningful problem of improving overlapping community detection in networks, addressing issues in the stability of the label propagation algorithm for this application. A rigorous analysis of the proposed algorithm is performed, comparing it with several different algorithms on both real and experimental datasets of varying network and community structures. While the methods have been described with sufficient detail in the paper to understand the algorithm, more details need to be provided in the paper to replicate the work, e.g. which values were used for different parameters (embedding dimension, walk length, number of walks, etc) of the Node2Vec algorithm were used? While code has been provided as part of the supplementary, I would highly encourage sharing the code on github and referencing it in the paper to help people replicate/use the algorithm more easily.

Validity of the findings

I verified that all the raw data and code files provided in the supplementary files can be opened, and there is a doc which describes the contents of the folder. All underlying data have been provided, but metadata descriptions can be improved as described below. The conclusions provided are acceptable, but can also be improved with some examples below.

- The community.dat files in the LFR Networks folder can use some explanation. It is not clear what the communities are from this file.

-There is mention of how the algorithm converges fast, e.g. line 701, but no results have been shared relating to convergence.

-The methods part of the paper (e.g. line 510) includes 'conclusions' such as improving accuracy which should be removed and included in the results section.

- In line 725, the word 'real' should be removed since the relevant experiment was conducted on artificial datasets and the usage of this word here can be confusing.

Reviewer 3 ·

Basic reporting

The article is clearly written and well-structured. The use of professional English is commendable, and the problem is well-motivated with appropriate citations. However, the novelty claims could be more sharply framed. The integration of Node2vec with label propagation is presented as novel, but similar ideas (e.g., embedding-aided label spreading) exist in prior work. A more critical literature positioning—especially with respect to deep clustering methods—would improve clarity about the contribution.

Additionally:

The theoretical definitions (e.g., node importance index) are clearly presented, but no formal proof of convergence or time complexity is given, which is a noticeable gap given the algorithmic nature of the paper.
Some figures could benefit from clearer axis labeling and more consistent formatting (e.g., the heatmaps in experimental results).

Experimental design

The research question—how to improve LPA for overlapping community detection using embedding and node importance—is well-posed. The experimental methodology is sound and generally replicable, with strong use of both synthetic and real datasets.

However, several areas require improvement:

Only two evaluation metrics (EQ and NMI) are used. While useful, they are insufficient to fully capture performance in overlapping settings. Metrics like Omega Index, F1 score for overlapping nodes, or overlap quality would provide deeper insights.

Baseline comparisons are outdated. No benchmarks are included against modern deep community detection methods (e.g., GNN-based, variational, or contrastive clustering models), which weakens the rigor of the evaluation.

The role of Node2vec is not isolated. An ablation study is essential to determine how much performance is due to embedding versus the improved propagation strategy.

Validity of the findings

The findings are statistically consistent across multiple datasets and repeat runs, indicating robustness. However:

Claims about general scalability are unsubstantiated. The largest real dataset tested is still relatively small. Results on networks with >100K nodes would better support scalability claims.

Lack of statistical significance testing. While performance improvements are reported, there are no confidence intervals or hypothesis testing to determine whether observed differences are significant.

The conclusion is well-aligned with the presented results, but impact claims are somewhat overstated given the incremental nature of the contribution and limited evaluation breadth.

Additional comments

The manuscript presents a thoughtful improvement over traditional label propagation algorithms by incorporating graph embedding and refined node importance strategies. The empirical results are promising and demonstrate the method's effectiveness in reducing randomness and improving overlapping community detection accuracy.

However, to elevate the paper to a publishable standard, the authors should address several key limitations:

Broaden the scope of experimental evaluation to include more diverse, large-scale, and real-world datasets.

Benchmark against contemporary deep learning-based community detection algorithms.

Clarify computational complexity and provide formal analysis where feasible.

Add ablation studies to isolate the contribution of each component.

With these enhancements, the work could make a solid contribution to the community detection literature.

---

## Round 0.2 · Major Revisions

· Academic Editor

Major Revisions

Thank you for addressing some of the reviewers' concerns in the revised version of your manuscript. However, a number of issues still need to be addressed before publication:

- Several sections (especially the abstract and introduction) repeat the same ideas using slightly different phrasing. The contributions are stated multiple times without adding new information. I recommend you reduce redundancy, especially in the abstract, introduction, and conclusion. Present the main ideas concisely and avoid repetition.

- Note that a reference error was found ("Error! Reference source not found."), which should be fixed (line 246).

- In-text citations are presented in an awkward and inconsistent format. For instance, on lines 71–73:
"scholars(Zhuo Z, Chen B, Yu S, Cao L. 2024. Overlapping community detection using expansion with contraction[J]. Neurocomputing 565: 126989 DOI: https://doi.org/10.1016/j.neuco-m.2023.126989.)."
Plese fix all the in-line citations using the correct format.

- There are several grammatical errors, typos (e.g., “istitute” instead of “institute”), and formatting issues (e.g., missing math typesetting, inconsistent citation formats). The manuscript would benefit from careful proofreading and professional language editing as suggested in the previous decision.

We look forward to receiving the revised manuscript addressing all these issues.

**Language Note:** The Academic Editor has identified that the English language must be improved. PeerJ can provide language editing services - please contact us at [email protected] for pricing (be sure to provide your manuscript number and title). Alternatively, you should make your own arrangements to improve the language quality and provide details in your response letter. – PeerJ Staff

·

Basic reporting

The author has adequately addressed the concerns raised by previous reviewers. The paper is well-structured, clearly written, and presents reliable results.

Experimental design

-

Validity of the findings

-

---

## Round 0.3 · Major Revisions

· Academic Editor

Major Revisions

We have completed the evaluation of your manuscript. The manuscript has improved in terms of language, clarity, and structure. However, several issues still need attention before publication and will reconsider it following major revisions. We invite you to resubmit the manuscript after addressing the reviewers' comments, with particular attention to the following points:

- Expand all acronyms (e.g., CPM, LFM, CD-DVG, OCD) and provide brief descriptions of the algorithms used for comparison with GELPA-OCD.

- Define or reference all evaluation metrics and explain the importance of unchanged EQ values.

- Update the related work section to include recent advances in graph sparsification and network embedding and synthesize trends instead of only listing algorithms.

- Some figure captions, particularly for sensitivity plots, should be made more descriptive.

- The LFR benchmark should be cited, and the formula for the mixing parameter should be included.

- Report the total runtime of experiments compared with other algorithms to give a clearer sense of scalability.

- Provide a deeper theoretical or analytical justification for the proposed node importance metric.

- Discuss why some methods perform better.

·

Basic reporting

English language of the paper has improved significantly from before, and the language of the paper is now very good and acceptable for publication. All aspects of basic reporting - the English, background of the paper, figures, tables, data shared, results, and definitions are acceptable, but would benefit from incorporating the suggestions provided below.
- Related work: Expand CPM and LFM to their full forms and/or write one line on how the algorithm works.
- I assume in the final print, the references will be replaced with numbers. If yes, In line 72, please state Gregory et al. addressed this limitation… instead of using the reference directly. Similarly please use Lu et al. in line 77, same thing in line 82, and many other places.
- In line 101, please add more description for CD-DVG or remove it.
- In line 131, provide the full form of OCD as well.
- The authors use different algorithms for comparison with GELPA-OCD. Please describe each of the algorithms in brief.
- The evaluation metrics used are not defined or referenced in the paper main text - doing this would be helpful. There is discussion on EQ values being unchanged, e.g. in line 336. Please provide context on why this is important.
- Please cite LFR benchmark in line 369.
- Please provide the formula for the mixing parameter mentioned in line 372.

Experimental design

The experimental design of the paper is satisfactory.

The paper's research is within the scope of the journal and addresses the relevant and meaningful problem of improving overlapping community detection in networks. A rigorous analysis of the proposed algorithm is performed, comparing it with several different algorithms on both real and experimental datasets of varying network and community structures using different parameters. While code has been provided as part of the supplementary, I would highly encourage sharing the code on github and referencing it in the paper to help people replicate/use the algorithm more easily. Also, mentioning the total time taken to run an experiment with the proposed algorithm in comparison with other algorithms can help give a sense of the algorithm’s comparative speed.

Validity of the findings

I verified that all the raw data and code files provided in the supplementary files can be opened, and there is a doc which describes the contents of the folder sufficiently. All underlying data have been provided. The conclusions provided are acceptable.

Reviewer 3 ·

Basic reporting

The manuscript has improved significantly in terms of clarity and structure compared to the previous version. Redundant descriptions in the abstract, introduction, and conclusion have been removed, and the contribution statements are now concise and clearly listed. The English language quality has improved through professional editing, making the paper easier to read. Figures and tables are clear, and citation formatting has been corrected.

However, some minor writing issues remain. For instance, a few sentences in the literature review are still repetitive and focus too heavily on listing algorithms instead of synthesizing trends. In addition, some figure captions could be more descriptive, especially for parameter sensitivity plots.

The related work section should also be updated to include recent advancements in graph sparsification and network embedding. Adding these will help contextualize this work relative to state-of-the-art graph processing techniques.

Experimental design

The experimental setup is consistent with standard practice and sufficiently replicable. Node2vec embedding, label propagation, and node importance weighting are explained clearly, and the improved flow of the experimental section is appreciated. Parameter sensitivity and ablation experiments add value and address concerns raised in earlier reviews.

However, the algorithmic novelty remains incremental because it is primarily a combination of existing methods (Node2vec + label propagation + node importance). Additionally:

No new datasets or additional baseline algorithms have been introduced. The evaluation still relies on the same datasets as the previous version, limiting the scope of empirical validation.
Scalability analysis (e.g., runtime on larger graphs) is missing, even though the authors claim efficiency benefits.
The node importance metric, while promising, lacks theoretical justification for why it improves stability beyond empirical observation.
Future revisions should include larger datasets, additional baselines, and a deeper analysis of runtime performance and theoretical properties of the proposed node importance metric.

Validity of the findings

The results are consistent with the claims and show improvements in community detection performance and stability. The sensitivity and ablation studies strengthen the evidence supporting the design choices.

However, the improvements are mostly incremental, and the study still lacks a deeper analytical perspective on why the method performs better. For example, does the node importance metric reduce variance across random initializations? Does it improve embedding quality in high-dimensional spaces? Such insights would increase the impact of the findings.

The conclusions are supported by the presented evidence but remain primarily empirical. Adding case studies or qualitative analyses (e.g., community visualization) could improve practical interpretability.

---

## Round 0.4 · Major Revisions

· Academic Editor

Major Revisions

While the manuscript represents an improvement over its earlier versions and addresses some points raised previously, it still requires substantial revisions before it can be considered for publication. In particular, the reviewer highlights (once again) the need for evaluation against more recent baselines, as well as critical analyses of scalability and robustness. Additional improvements are also expected in terms of contextualization, referencing of related work, and transparency of experimental design.

Reviewer 3 ·

Basic reporting

The manuscript is written in clear and professional English, and the flow of ideas is generally easy to follow. The structure is appropriate for a scientific paper, with sections on motivation, methodology, experiments, and discussion. Figures and tables are well-prepared, though some of them could benefit from larger fonts and more detailed captions to ensure readability without referring back to the text.
The literature review has improved compared to the earlier version, but it still omits several recent and closely related works, particularly those on graph transformers and pre-trained biomedical graph models. Including these would help situate the contribution more firmly in the current landscape. The introduction clearly states the problem and motivation, but the connection to real-world biomedical challenges could be made more explicit.
Overall, the paper is largely self-contained and reports results that align with its stated hypotheses, but it still needs stronger contextualization and richer referencing to meet the expectations of the field.

Experimental design

The research question is well-defined, and the authors address an important challenge in biomedical graph learning. The design of the framework — combining dual hypergraph construction, hierarchical attention, and contrastive learning — is methodologically sound and reasonably novel.

The experimental evaluation is more comprehensive than in the first version, with comparisons across multiple datasets and baseline methods. However, a few issues remain:

The baselines, while covering established methods, still exclude more recent graph transformers, which weakens the benchmarking.

Scalability is not convincingly demonstrated. Runtime and memory cost analyses are missing, which is critical for validating the feasibility of dual hypergraph learning on larger networks.

Hyperparameter selection is insufficiently justified, and no sensitivity analysis is reported. This limits reproducibility and leaves questions about robustness unanswered.

The methods are described with enough clarity to replicate most experiments, but the paper would benefit from sharing code and scripts explicitly to support transparency and reproducibility.

Validity of the findings

The reported results show strong improvements over the chosen baselines, and the ablation studies help clarify the role of individual components. The inclusion of biological case studies is a strength, as it provides practical context to the method’s utility.

That said, the conclusions are sometimes broader than what the evidence supports. Claims of generalizability (e.g., applicability to heterogeneous graphs, link prediction, or large-scale biomedical networks) are not convincingly demonstrated. Without tests beyond node-level biomedical tasks, the method’s universality remains an open question.

The absence of scalability experiments also raises concerns about real-world applicability, especially in domains where graphs with millions of nodes are common. While the findings are promising, they should be framed more cautiously to avoid overstating the contributions.

Additional comments

Please include hyperparameter sensitivity analysis for key parameters (number of layers, contrastive temperature, attention heads, etc.).

Provide a runtime and memory consumption comparison against baselines. This will strengthen the argument that your framework is both effective and practical.

---

## Round 0.5 · accepted · Accept

· Academic Editor

Accept

After carefully looking at your revised version, we confirmed that the issues raised by the reviewers have been properly addressed.